# Does a Kegel Exercise Program Prior to Resistance Training Reduce the Risk of Stress Urinary Incontinence?

**DOI:** 10.3390/ijerph20021481

**Published:** 2023-01-13

**Authors:** Donelle Cross, Marilynne N. Kirshbaum, Lolita Wikander, Jing-Yu (Benjamin) Tan, Simon Moss, Daniel Gahreman

**Affiliations:** 1Faculty of Health, Charles Darwin University, Darwin, NT 0909, Australia; 2College of Nursing and Health Sciences, Flinders University, Bedford Park, SA 5042, Australia; 3Research and Innovation, Charles Darwin University, Darwin, NT 0909, Australia; 4Department of Sport, Exercise, Recreation and Kinesiology, East Tennessee State University, Johnson City, TN 37614, USA

**Keywords:** stress urinary incontinence, resistance training, Kegel, women’s health, pelvic floor

## Abstract

This comparative pre–post intervention study investigated the feasibility and benefits of Kegel exercises amongst incontinent women, prior to commencing resistance training (RT), to reduce the risk of stress urinary incontinence (SUI) compared to a group of women without prior Kegel exercises (KE). Incontinence severity index (ISI) score, pelvic floor muscle strength (PFMS), and body composition (such as body mass index (BMI), fat, and muscle mass), were obtained pre and post intervention. Results demonstrated that RT reduced SUI to a significantly greater extent only if preceded by KE as was observed in the Kegel exercise plus RT group (KE + RT) over time. The improvements in total ISI in both the KE + RT and RT groups were large (d = 1.50 and d = 1.17 respectively). A two-way ANOVA indicated a statistically significant improvement in average PFMS within the KE + RT group over time and between the two groups. A positive correlation was found between the average strength of pelvic floor muscles and SUI. Participants in KE + RT group demonstrated a significant increase in muscle mass (*p* ≤ 0.001) and concomitant reduction in fat mass (*p* = 0.018). This study determined a dedicated program of KE preceding a RT program improved average pelvic floor muscle strength and was effective in reducing SUI among incontinent women.

## 1. Introduction

Urinary incontinence affects up to 70% of women worldwide [1] with stress urinary incontinence (SUI) reported as the most prevalent sub-type [2]. One study determined 40% of women in the U.S. experience SUI [3], a study in China considered almost 34% of SUI in a cohort of 4000 women to be high [4] and in a systematic review, prevalence of SUI varied from 12.5% up to 79% [5]. SUI is debilitating and can inextricably restrict lifestyle, reduce quality of life, and lead to social isolation [6], and/or having to modify or avoid sport completely [7]. The definition of SUI is the involuntary loss of urine on effort or physical exertion excluding sporting activities, or on sneezing or coughing [8]; and can be provoked by resistance training (RT). RT includes exercises using free weights and/or machines where an individual uses their own muscles against the weighted resistance; this ‘resistance’ increases over time with increased strength [9]. The leakage of urine, especially in response to performing RT, can discourage women from engaging in regular RT which has been shown to have positive effects on health and wellbeing [10].

Contrary to previous reports, some incontinent women who continued to perform RT experienced an improvement in their daily continence [11]. Physically active women tend to have stronger pelvic floors and this is thought to contribute to a successful continence mechanism when there is an increase in their intra-abdominal pressure [12]. Indeed, the results of a recent study found over 8% of the incontinent women no longer experienced urinary leakage during their daily life after performing RT [11]. This phenomenon may suggest that RT could have a positive impact on pelvic floor strength and women’s ability to timely and effectively activate their pelvic floor muscles [11].

A body pump program which included squats, deadlifts, and bench press among other activities, concluded that general RT does not cause urinary incontinence [13]. However, the results of another study demonstrated that many women who engaged in Crossfit and Powerlifting, with similar activities, did experience incidences of urinary incontinence; the authors found that some of the participants of these studies were unsure whether they were activating their pelvic floor correctly [14], and this may have led to experiencing urinary incontinence. Activating the ‘knack’, an element of the Kegel exercise, to counterbrace the pelvic floor at the time of a focused deadlift or squat when intra-abdominal pressure (IAP) is momentarily increased, is thought to be enough to facilitate urethral closure and subsequently prevent leakage [15].

Pelvic floor muscles are vital to maintaining continence. A deficit in urethral closure is the cause of leakage in SUI; this can result from weak pelvic floor muscles unable to lift the bladder or perform effectively to close the urethra [16]. Kegel exercises are established as an effective modality to reduce SUI [17] and remain the gold standard for first-line treatment [18]. The benefit of Kegel exercises has been supported in many studies since Arnold Kegel’s concept [19,20] where an improvement of 84% was seen in his initial studies [21]. In fact, the efficacy of reducing stress urinary incontinence with Kegel exercises has been cited as having success rates from 27% [22] to satisfaction rates cited up to 86% [23] in another study.

It is unfortunate that many sports are avoided due to SUI. In agreeance with a similar thought process related to continence threshold in another study [14], it is offered that the provocation of urinary incontinence in RT occurs as the result of inadequate strength of pelvic floor muscles, or the inability to correctly and timely activate them to counter the increase in intraabdominal pressure that surpasses a woman’s individual continence capacity. A systematic review found that exercises which focused on the participant maintaining an awareness of PF muscle contractions, improved the efficacy of the activity [24]; this aligns with the results of our study.

Despite the increasing popularity of RT among women, a pre-exercise screening does not exist that includes information regarding pelvic floor muscles health or women’s ability to effectively activate their pelvic floor muscles during training. Knowledge and the ability to effectively activate pelvic floor muscles, in addition to adequate coaching skills for proper set up and bracing, may reduce the incidence of SUI during exercise. Hence, this study aimed to determine whether a program of Kegel exercises prior to a RT program will result in reduction in SUI and should be prescribed to incontinent women prior to performing RT. We hypothesized that performing KE prior to RT would improve pelvic floor muscle strength (PFMS) and reduce the odds of experiencing SUI during RT.

## 2. Materials and Methods

### 2.1. Study Design

The data used in this study for the RT group were from a foundational single cohort that investigated the impact of RT on SUI in a group of women participating in a RT program who had not previously performed Kegel exercises. New data for the KE + RT group were from two consecutive studies where the participants all completed a program of Kegel exercises (supervised vs. unsupervised) with the results already published [17]. These 19 participants from the second study were invited to partake in 12 weeks of RT to compare against the RT group. Figure 1 summarizes the study protocol of the two separate groups of women compared in this study.

### 2.2. Study Sample

This report compares the findings of two separate studies, as described in Figure 1, in which the effect of RT on SUI and PFMS was compared between participants who completed 12 weeks of RT with or without prior Kegel exercises. The RT group consisted of 14 incontinent women who did not have any experience in RT or Kegel exercises and completed 12 week of RT. Participants in the Kegel exercise plus RT (KE + RT) group (*n* = 19) had completed 12 weeks of Kegel exercises under the supervision of a qualified pelvic floor specialist [17]; they then completed the identical RT program as the RT group. A convenience sample was used in this study due to time requirements and availability of space and resources to carry out the project.

### 2.3. Study Interventions

Pelvic Floor Assessments and Resistance Training (RT). Each participant met with a women’s health physiotherapist for an initial one-hour assessment where data collection of obstetric/gynaecological history, specific triggers for SUI (for example sneezing, exercise, etc.), safety for inclusion into the RT program was ascertained and written and verbal consent obtained for the vaginal pelvic floor (PF) examination. A consistent process using agreed perineometry and digital palpation was used for all participants.

Perineometry is often used by physiotherapists as a validated technique [25]. Therefore, in this study, pelvic floor muscle strength (PFMS) was measured using a perineometer (“Peritron™”, Cardio-Design, Australia). The silicone rubber sensor was sheathed with a lubricated latex-free condom and inserted vaginally to detect a resting baseline measured in cm/H_2_O of PFMS; this did not entail squeezing. To allow an ‘average’ measurement of PFMS contraction to be calculated, five contracting measurements were taken with the maximum number registering after 5 s of squeezing on the sensor reported. A 30-s rest interval between measurements was implemented to avoid fatigue and allow adequate time for recovery. Lastly, an 80% maximum voluntary contraction (MVC) measurement from previous attempts were recorded and participants were asked to maintain this contraction for up to 20 s. For consistency during measurements, instructions of ‘squeeze and lift’ and ‘hold’ (the contraction for example), were verbalized. Co-contractions of abdominal, gluteal, or hips muscles were discouraged during the process. As a tool that has proven reliability and valid [26], the Peritron^TM^ was selected for this study ensuring consistency, reliability of data and fidelity of intervention.

Currently there is no one, commercially available tool that can provide a comprehensive assessment of the strength and function of the pelvic floor group of muscles; therefore, we have used the current standard instruments of adjunctive digital palpation, concomitant with Perineometry [27]. During the examination, the participant was asked to lie down with two pillows elevating their head; the position allowed the physiotherapist to assess more easily for tone, atrophy, and the contraction/relaxation of the pelvic floor muscles. Post study pelvic floor assessment was identical perineometry and digital palpation assessment but was only thirty minutes duration as it did not include any health history collation.

The RT program consisted of back squats, deadlifts, incline pull-ups, and push-ups on knees, and was based on the Powerlifting Australia Level 1 Powerlifting Coaching course. It was adapted to a two day/week program with a focus on squats and deadlifts which were chosen as they both require significant amount of pelvic floor engagement to counteract the IAP created when bracing in preparation for a repetition. One-hour sessions were scheduled with the expectation of attendance twice per week to a maximum of twenty-four sessions. Compliance was managed with flexibility of multiple sessions to choose from, and attendance was documented by trainers along with notes made each session for continuation of training in next session. To ensure safety of participants with correct lifting technique, classes were limited to a maximum of six participants.

Each lift was performed in training twice a week; one day as a heavy lift day and one day as a light day focusing on speed. Two days were selected as it was determined that to be enough to elicit a change in strength and easier to adhere to than a three day a week program. The program was based on a percentage of the participants predicted one repetition maximum (1-RM) lifts; the repetitions then became progressively heavier as repetitions became fewer.

Sessions commenced with stretches and warm up activities to facilitate flexibility of large muscle groups (hamstrings and quadriceps), and hip movement, in addition to 2–3 sets of squats and deadlifts with progressive loads. All participants commenced squats using a broomstick. They squatted to progressively lower and lower boxes until they could squat for 3 × 10 sets to a depth where the hip crease was lower than the top of their knee. Once they achieved this, they progressed to 10 kg, 15 kg, and then 20 kg bar before being loaded and programmed. Table 1 highlights an example of a training schedule once a participant was at the programmed stage.

The participants all started with a 25 kg deadlift with weights increasing once they could complete 3 × 10 sets of repetitions with safe form. The aim was to increase load incrementally as tolerated at 2.5 kg–5 kg per week. Once the participants had a regular program, training alternated between heavy lifts/light squats one day and then light lifts/heavy squats then next. Rest periods between sets of repetitions was enforced and ranged from five minutes in the early stages, up to twelve minutes in the final weeks with the increasing weights. As training progressed, repetitions reduced as the weights/intensity increased; this was specific to the individual participant’s capabilities.

Coaches trained participants to breathe and brace correctly using the pebble analogy as a visual explanation of how to activate the pelvic floor during a lift/squat. They were asked to visualize ripples from a pebble dropping into a pond, and then to rewind the image of the ripples becoming smaller and the pebble lifting out of the water. It was emphasized this was to be performed with each lift and to stop if they lost connection with their pelvic floor muscles due to fatigue. Verbal reminders were provided throughout training sessions

After each training session, push-ups, cool down stretches, and foam rolling exercises were completed. Participants were asked at each session about their pelvic floor health/urinary incontinence status to ensure that the weight training was not proving deleterious in any way. All participants were trained by qualified accredited powerlifting coaches

### 2.4. Study Procedure

Both the RT group and KE + RT group were recruited following identical recruitment procedures started with an expression of interest to flyers advertising the project in local shopping centers, social media and around a university campus. All participants in this study met the criteria of being female, over eighteen years of age, experience SUI, did not perform Kegel exercises, had no experience in regular resistance/training, and were not currently pregnant or breastfeeding for the duration of the study. All participants were informed about the conditions of the program including the requirement for a pelvic floor assessment, an outline of the RT program and the commitment required of 12 weeks including at least 80% attendance at the training session. A self-screening tool “physical activity readiness questionnaire (PAR-Q)” was required to rule out participants not suitable for physical exercise. They were also informed they could withdraw from the study at any point. Written consent was received from all participants.

### 2.5. Outcome Measures

Strength of pelvic floor muscles was measured by a female physiotherapist and reported in cm/H_2_O. Degree of urinary incontinence was evaluated from a self-reported questionnaire, the Incontinence Severity Index (ISI), which was completed prior to the RT program and at the end of the program. The ISI assesses the degree of incontinence and includes frequency of urine leakage and severity of symptoms and is a validated tool considered to provide excellent results [28].

### 2.6. Other Assessment

A Registered Nurse (RN) measured heart rate and blood pressure for each participant and collected additional data of height, weight, and body composition analysis.

### 2.7. Statistical Analysis

Data were analyzed using IBM Statistical Package for the Social Sciences (SPSS) version 26 (New York, NY, USA). To analyze the difference between the RT group and the KE + RT group, a series of two-way ANOVAs were performed with the α value set to 0.05 to identify statistical significance. The effect of the intervention was determined using Cohen’s d formula, and Pearson correlations were calculated to explore relationships between the factors

## 3. Results

A total of 24 participants aged between 27 and 62 years of age completed the study with an average age of 49.7 years. Two participants in the KE + RT group withdrew during the RT program, one due to exacerbation of arthritis, and the other declined the final PF assessment due to COVID-19 restrictions. Data from five participants have been removed from the RT group and two further from the KE + RT group as they did not meet the requirement of attending 80% of the RT sessions. The analysis in this study was based on comparable data collected from 9 participants in the RT group and 15 participants in the KE + RT group. Participants were not randomized into groups in this study but were individuals who had been recruited at different times however, recruitment and eligibility criteria were the same in each group. Table 2 summarizes the mean and effect size of variables between the two groups before and after the intervention.

The frequency, severity, and overall SUI significantly reduced over time for both the RT group and the KE + RT group. This improvement was especially pronounced in the KE + RT group. An insignificant reduction in MVC was observed in the RT group with a moderate positive increase in the KE + RT group.

A two-way ANOVA (see Table 3) assessed whether the impact of the intervention on the various measures differed between the two groups. The improvements in total ISI and frequency of urine over time were both found to be statistically significant. The improvement in frequency, severity of symptoms, and ISI of urinary incontinence symptoms over time was also significant. However, in this instance, the interaction between group and time was not significant, indicating this improvement in frequency, severity, and ISI score of urinary incontinence did not depend on whether Kegel exercises preceded the RT.

Average pelvic floor muscle strength also increased over time. The interaction between group and time was not significant although the impact of time and effect of Kegel exercises preceding RT was statistically significant (both *p* ≤ 0.001). Conversely, MVC decreased over time in the RT group but increased over time in the KE + RT group—and again this interaction between group and time was significant. A Pearson correlation where 0 indicates no correlation and −1 indicates a perfect negative correlation, detected a statistically significant negative correlation between stress urinary incontinence and pelvic floor muscle strength, r(24) = −0.564, *p* = 0.004.

A statistically significant increase in muscle mass and decrease in fat mass was observed over time (*p* = 0.001 and *p* = 0.018 respectively); again, these changes did not differ between the groups. A statistically significant improvement was noted in weight and BMI over time however there was no significant interaction observed between the two groups and the intervention. A Pearson correlation determined a statistically significant correlation between the reduction in fat mass and increase in muscle mass r(30) = −0.709, *p* < 0.001, indicating a strong inverse relationship between these two factors.

No significant correlation was found on analysis of the relationship of the PFMS and ISI when exploring changes. Similarly, when considering results prior to the intervention and after the intervention, the relationship between these two was found to be insignificant.

## 4. Discussion

This study investigated whether performing KE prior to RT would improve pelvic floor muscle strength (PFMS) and reduce the odds of experiencing SUI during RT. We were also interested in capturing the additional benefits in improving pelvic floor strength and reducing urinary incontinence.

The results of this study supported the hypothesis and demonstrated that participants in the KE + RT group experienced a greater reduction in UI. This study confirms the importance of Kegel exercises as a significant factor in reducing SUI with a substantial improvement noted in the reduction in the overall ISI score in the KE + RT group. No studies were found in the literature of the efficacy of Kegel exercises prior to a RT program as being beneficial as such; this is the first study of its kind.

Surprisingly, it was noted that the exercise of RT was also effective in reducing SUI. This improvement in SUI amongst the participants could be attributed to practicing appropriate bracing technique during lifting sessions, and adequate supervision of these sessions under a qualified coach who emphasized on their technique and ability to effectively contract PF muscles, then the amount of weight lifted.

A similar study, though with a cohort of women over 60 years of age, was found comparing combined weight training with PF muscle training versus PF muscle training alone and found the combination of weight training (using bodybuilding machines) with PF exercises, improved SUI as indicated by absence of symptoms by over 58%, compared to almost 15% in the PF training alone in the first 4 week, indicating that weight training may have been a successful contributing factor [29]. It was suggested in that study that the increase in pelvic floor muscle strength, specifically the pubovisceral muscles from Kegel exercises, was associated with an earlier improvement noted in the combined Kegel and weight training group.

In another study of women aged 18–49 years, exploring PF muscle training versus PF muscle training and abdominal training (AT), similar to RT in activating the transverse abdominis (TrA) muscle, it was found that the addition of AT with PF muscle training contributed to an early improvement in SUI [30]. Neither of these papers provided further studies to determine whether a program of Kegel exercises prior to the activity of weight training or AT would be more effective as our study has shown. It is thought, in our study, that the improvement in pelvic floor muscle strength and SUI already reported as statistically significant in a previous study [17]; that the activities in the RT program have enhanced this strength reducing SUI further. This is supported in another study where it is suggested that the PF muscles could be strengthened further with physical exercise and the co-contraction of these muscle groups [31].

RT is not yet accepted as a strategy to improve pelvic floor muscles, but it appears that women who completed KE were able to activate their PFMs during RT sessions and hence enhanced the strength of PFMs, especially to counteract the increased intrabdominal pressure. Kegel exercises essentially work on the principle of overloading or training the muscle to replicate its functional movement [32], to provide support to the pelvic organs including the bladder. The importance of strong pelvic floor is vital to continence as demonstrated by the strong correlation of findings in this study between SUI and PF muscle strength. In our study the average measurement of strength of the PF muscle increased in both groups.

Our study detected an anomaly of sorts in the reduction in maximum voluntary contraction (MVC) in the RT group; however, an increase in strength was observed in the KE + RT group. Measurement of the MVC is taken during the pelvic floor contraction and is representative of activation of multiple muscle fibers; the more a fiber is activated, the stronger the contraction is seen/reported. Conversely, other studies have also identified an increase in MVC strength as linked to a reduction in urinary incontinence [33]. It could be suggested that RT without the benefit of preceding Kegel exercise program, could facilitate greater muscle fatigue as MVC pressure can reduce by up to 20% after strenuous exercise [34] and that Kegel exercises prior have produced a stronger PF muscle foundation. This is supported in other research where it was concluded that the inclusion of Kegel exercises in training programs of women practicing sport is recommended [35]; this aligns with results of our study which demonstrated an improvement in MCV strength in the KE + RT group.

The role of Kegel exercise as an isolated therapy has proven effective in treating SUI [17,36]; however, the benefit to include it as a dedicated program prior to physical exercise needs to be seriously considered. Although more trials are needed, preliminary evidence from this study demonstrate a statistically signification improvement in ISI scores with reductions in frequency and severity of SUI in both groups indicating that perhaps the RT program itself, may have been contributory. It is proposed that the actions involved in RT activities improved control of pelvic floor muscles, this is supported in the literature affirming the urinary system can be modified by behavioral and musculoskeletal factors, including through activities such as physical exercises [37].

General RT does not cause SUI with no significant changes detected in a group of women doing Body pump, which included deadlifts and squats, during an intervention period [13]. It is believed the intensity and focus on IAP during a squat or deadlift in our program, allowed the participant time to contract the PF muscle (the ‘knack’), in anticipation of the lift/squat. The TrA muscle is activated during RT, and when trained with pelvic floor muscle exercises, can produce significant results by closing the sphincter better thereby decreasing/eliminating urine leakage, in fact the TrA muscle was found to be activated 224% more when a maximum PF contraction was performed [38]; in later studies, these authors contend this synergistic relationship of abdominal muscles and PF can improve a woman’s ability to contract their PF muscles [39]. Evidence from a systematic review further advocated this finding of synergism between the PF and TrA [40].

When IAP increases beyond the continence threshold, SUI may be experienced. As a result, several sports/exercises which provoke a high IAP are not considered pelvic floor friendly, for example, muscle training sports, such as lifting weights. It is acknowledged that chronic IAP can affect the PF with excessive pressure [37]; however, it is suggested that deadlifts and squats, are not considered ‘chronic’ given the short, intermittent moments of IAP which are interspersed with resting time as was enforced in our study. RT with adequate rest intervals and correct bracing technique could indeed be beneficial to enhance the strength of PFMs and in turn reduce SUI. It is vital, however, to remember that increasing the lifted weight is directly associated with increasing IAP and should only happen when individuals’ PF muscles are strong enough to counteract the increased IAP resulting from greater weight.

Our study subsequently demonstrated that a program of RT can contribute to reducing SUI agreeing with the literature that stretching and fatiguing the PF muscles, in sports such as RT, provides the effect of indirect training of elevating the PF and reducing incidence of SUI [41]. It is thought that the improvement seen in the RT group was a result of the progressive weights being more intense than any internal squeezing of PF muscles in Kegel exercises could be, however the author contends that the improvement in MVC in the KE + RT group, was attributed to learned habit from the preceding Kegel exercise program of contracting PF muscles and the automatic activation of this action during lifts/squats resulting in a stronger contraction in this group. This aligns with findings in a scoping review which determined an instinctive habitual muscle action could develop over time as a consequence of learning the skill of deliberately activating the pelvic floor, such as in the action of Kegel exercise [42].

Low to moderate RT, with scheduled rest periods between focused lifts does not aggravate SUI. Previous researchers identified heavy weights and inadequate rest and fatigue occurring at the end of high repetitions sets as SUI provoking factors [43]. However, our findings contend that a program of Kegel exercise program prior, is beneficial to provide initial strength and possibly muscle memory to the PF muscles affording an initial stability prior to RT; this improvement or change in functional structure has been found to result from PF muscle training [44].

It was reassuring to note in our study, the intervention of using increasing strengths (such as weights) moderately with repetitive sets to build strength and muscle, showed a positive correlation between reduced SUI and improved physical fitness [29,45], recognizing the tacit health benefits of physical exercise. Our study demonstrated a similar reduction in fat mass observed in both groups with a corresponding increase in muscle mass reported, indicating the additional impact of regular physical exercise from a general health perspective. Further results established a significant and positive correlation between the concomitant reduction in fat mass and increase in muscle mass generally. A specific BMI was not a criterion in this study however needs consideration when commencing RT as IAP is already elevated in overweight women

## 5. Limitations

Despite the interesting and practical information presented in the results of this research, a few limitations were noted. Firstly, training programs were individualized and were progressed based on women’s ability to maintain PFMs contracted during a lift. In some cases, the number of repetitions and sets indicated in the RT program was not achieved due to PF muscle fatigue. Secondly, no standard training was performed between the multiple assessors to standardize practice despite clear process, form, and perineometry used. Whilst an identical process was followed, subjective assessment may have skewed results of average PF muscle strength in the RT group as the figures whilst consistent across those participants, were greatly different to the initial reading whereas the results for the KE + RT group, remained consistent with a small improvement. Finally, accurate causality assessment between the study interventions and the outcome measures may be using a cohort of participants from a previous study without paralleled group allocation and randomization may have impacted the results. It is acknowledged that the sample size was small and this may impact on the results, as a consequence; research evidence identified from this study can therefore be viewed as preliminary and should be examined further in a rigorously designed randomized controlled trial with power-based sample size estimation.

## 6. Conclusions

This study confirmed that supervised, progressive RT can be safely performed by incontinent women without exacerbating their UI if they have undergone a PFM assessment and developed skills to timely and effectively contract their PFMs during RT. However, the results of this study strongly supported the notion of pelvic floor assessments and supervised KE prior to performing RT. Participants in this study were reminded frequently to activate PF muscles as part of engaging their core, when attempting the RT exercises and it is suggested this PF focused supervision during training, contributed to the successful results. It is anticipated that results of this study will be of interest to coaches and trainers who will be encouraged to embed and advocate the importance of effective Kegel exercise programs prior to participation in RT.

## Figures and Tables

**Figure 1 ijerph-20-01481-f001:**
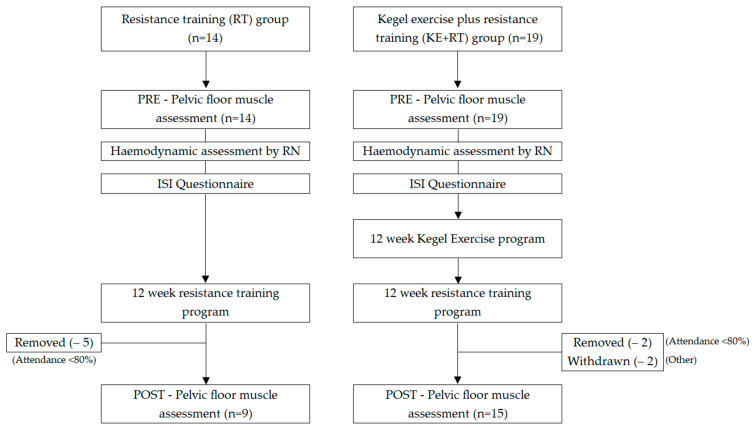
Summary of study protocol.

**Table 1 ijerph-20-01481-t001:** Example of a training program.

		Warm Up Reps	Sets	Deadlift (kg)	Squat (kg)	Rest Time (mins)
Lead up weeks		training specific to individual’s capabilities				
Week 4	Session 1	10, 10	3 × 10	40	27.5	8
	Session 2			25	22.5	8
Week 5	Session 1	10, 10	3 × 10	45	20	8
	Session 2			27.5	27.5	8
Week 6	Session 1	8, 5, 5	3 × 8	50	22.5	8
	Session 2			30	35	8
Week 7	Session 1	8, 5, 5	3 × 8	55	22.5	8
	Session 2			32.5	40	8
Week 8	Session 1	6, 5, 3	3 × 6	60	25	8
	Session 2			35	42.5	8
Week 9	Session 1	6, 5, 3	3 × 6	65	25	8
	Session 2			35	45	8
Week 10	Session 1	5, 3, 2	3 × 5	70	27.5	10
	Session 2			37.5	47.5	10
Week 11	Session 1	5, 3, 2	3 × 5	75	27.5	10
	Session 2			37.5	50	10
Week 12	Session 1	5, 3, 1, 1	2 × 2	85	27.5	12
	Session 2			37.5	60	12

**Table 2 ijerph-20-01481-t002:** Mean, standard error, and effect size of dependent variables in study.

	Resistance Training (RT) Group (*n* = 9)			Kegel Exercise + Resistance Training (KE + RT) Group (*n* = 15)		
	Pre	Post	ES	Pre	Post	ES
ISI Total score	2.89 ± 0.93	1.44 ± 0.73	1.73 *	4.53 ± 3.31	1.60 ± 1.50	1.14 *
- Frequency	2.33 ± 0.71	1.44 ± 0.73	1.24 *	2.60 ± 0.91	1.33 ± 0.91	1.34 *
- Severity	1.33 ± 0.50	1.00 ± 0.00	0.94 *	1.67 ± 0.72	1.00 ± 0.53	1.11 *
PF at rest (cm H_2_O)	31.17 ± 13.76	31.11 ± 10.24	0.005	32.71 ± 13.56	30.11 ± 8.03	0.23
PF_attempts (Average)	45.10 ± 20.71	66.32 ± 20.36	1.03 *	27.38 ± 12.00	38.35 ± 12.65	0.89 *
MVC @80%	18.33 ± 5.00	17.33 ± 5.39	0.19	14.00 ± 6.78	16.93 ± 5.38	0.48
BMI kg/m^2^	28.31 ± 4.64	28.62 ± 4.67	0.07	25.28 ± 4.88	25.74 ± 5.18	0.09
Fat Mass kg	31.79 ± 11.47	30.20 ± 10.45	0.14	23.47 ± 8.95	22.23 ± 9.18	0.14
Muscle Mass kg	43.79 ± 4.16	46.19 ± 3.19	0.65 *	41.69 ± 3.65	44.00 ± 4.08	0.60 *

Note: Cohen’s d value effect size based on d = small 0.2, medium effect 0.5, and large effect 0.8. Abbreviations: ISI—Incontinence Severity Index, PF—pelvic floor, MVC—maximum voluntary contraction time held in seconds, BMI—body mass index. * Significant difference from pre-intervention.

**Table 3 ijerph-20-01481-t003:** Two-way ANOVA results of difference between the RT group and KE + RT group.

		df	Mean Square	F	Sig.	Partial ETA Squared
URINARY INCONTINENCE FACTORS						
UI: Frequency	Interaction	1, 22	0.4	1.8	1.94	0.76
	Condition	1, 22	0.07	0.05	0.82	0
	Time effect	1, 22	13.07	58.54	0.00 *	0.73
UI: Severity	Interaction	1, 22	0.31	1.47	0.24	0.06
	Condition	1, 22	0.31	0.79	0.38	0.03
	Time effect	1, 22	2.81	13.26	0.00 *	0.38
Total ISI Score (frequency*severity)	Interaction	1, 22	6.24	3	0.1	0.12
	Condition	1,22	9.11	1.33	0.26	0.57
	Time effect	1, 22	53.9	26.02	0.00 *	0.54
Average PF muscle strength	Interaction	1, 22	295.89	2.84	0.11	0.11
	Condition	1, 22	5871.71	14.83	0.00 *	0.4
	Time effect	1, 22	2913.86	27.99	0.00 *	0.56
PF resting pressure	Interaction	1, 22	243.54	4.54	0.04 *	0.14
	Condition	1, 22	0.62	0	0.96	0
	Time effect	1, 22	37.04	0.69	0.41	0.02
MVC of PF at 80%	Interaction	1, 22	43.51	2.06	0.17	0.85
	Condition	1, 22	63.01	1.36	0.26	0.06
	Time effect	1, 22	10.51	0.5	0.49	0.02
PHYSICAL HEALTH						
Weight	Interaction	1, 22	0.24	0.1	0.76	0.01
	Condition	1, 22	1220.7	3.69	0.07	0.14
	Time effect	1, 22	13.64	5.72	0.03 *	0.21
BMI	Interaction	1, 22	0.02	0.18	0.68	0.01
	Condition	1, 22	98.35	2.06	0.17	0.09
	Time effect	1, 22	1.67	4.73	0.04 *	0.18
Muscle Mass (kg)	Interaction	1, 22	0.03	0.01	0.91	0
	Condition	1, 22	51.63	1.89	0.18	0.08
	Time effect	1, 22	62.31	36.14	0.00 *	0.62
Fat Mass (kg)	Interaction	1, 22	0.34	0.09	0.77	0
	Condition	1, 22	746.85	3.97	0.06	0.15
	Time effect	1, 22	22.51	5.86	0.02 *	0.21

Note: Interaction = time*group, Condition = between subject effects. Abbreviations: UI—urinary incontinence, ISI—Incontinence Severity Index, PF—pelvic floor, MVC—maximum voluntary contraction, BMI—body mass index. * Indicates a statistically significant effect.

## Data Availability

Not applicable.

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
