# Peer review of "Does a Kegel Exercise Program Prior to Resistance Training Reduce the Risk of Stress Urinary Incontinence?"

_ijerph, 2023, doi:10.3390/ijerph20021481_

Round 1

Reviewer 1 Report

dear authors, the manuscript covers an interesting topic and is well written. however, please see my comments/suggestions below to further improve the quality of the paper

Abstract: Please mention the type of study in the methods. 

Methods: Please describe the inclusion and exclusion criteria. 

Please explain how the participants were recruited

Please describe the sample size calculation?? is this sample size enough to generalise the results?

Results: What were the patient characteristics/demographics data? was there any relation or impact? 

because there is a different intensity of kegel excercise based on patient age, body weight (BMI) and other functional status (mobile vs bed ridden). please explain this in results. 

Please include the comorbidities and medication history data. 

were the patients with UI taking any medication?

all the best

Author Response

Dear Reviewer 1 - Thank you for the time you have taken to review our article. Your expertise and feedback is greatly appreciated and we believe that we have addressed this your feedback and improved on this article appropriately for inclusion into this quality journal.

Point 1:       Abstract: Please mention the type of study in the methods. 

Response1: Thank you. We have amended this; you will note the addition of - “This comparative pre-post intervention study…..“ to address your review. [see line 14 of the tracked version of the manuscript].

Point 2:       Methods: Please describe the inclusion and exclusion criteria. 

Response2: Thank you. Please note that information about the inclusion and exclusion criteria is described under the Study Procedure heading [lines 192-195 of tracked version of the manuscript].

Point 3:       Please explain how the participants were recruited

Response3: Thank you. Please note that information about participant recruitment is highlighted under Study Procedure heading [lines 200-205 of tracked version of the manuscript]

Point 4:       Please describe the sample size calculation?? is this sample size enough to generalise the results?

Response4: Thank you. We have added the following – “A convenience sample was used in this study due to time requirements and availability of space and resources to carry out the project”. [line 117-118 of tracked version of the manuscript].

Point5:        Results: What were the patient characteristics/demographics data? was there any relation or impact? 

Response5: Thank you. We have included the following

                   “A total of 24 participants aged between 27 and 62 years of age completed the study with an average age of 49.7 years”  [see lines 228-229 of tracked version of the manuscript]

Point6:        because there is a different intensity of kegel exercise based on patient age, body weight (BMI) and other functional status (mobile vs bed ridden). please explain this in results. 

Response6: thank you. Please note that “patients” were not a part of this study, nor any person that was bedridden. The study included a cohort of healthy women in a non-clinical setting designed to inform practice for women in the general population with a similar profile. We maintain that the variables of intensity of the Kegel exercises, body weight, BMI and functional status would also vary in the real world. 

Point7:        Please include the comorbidities and medication history data.  

Response7: Thank you however we feel this is not applicable to this study – this data was not collected – the participants were not “patients”, therefore this was not a consideration as it was a study of otherwise healthy women.

Point8:        were the patients with UI taking any medication?

Response8:      No – medication history was not collected. Participants all had some degree of SUI but were not medicated for this condition.

Again - I humbly thank you for your feedback and expertise. We believe we have made a number of positive changes to this article and believe it to be of the quality reflected as those from this journal.

Author Response

Dear Reviewer 2 - Thank you.  We appreciate the time you have taken to review this paper and hope that we have addressed all of your concerns.  We look forward to a favourable response from you.

Point1:        Summary The abstract summarizes the article.
However, the first two sentences in lines 14 and 15 can form single sentence. 
More and clearer information about the methodology can be provided (eg. what is meant by body composition in the line 18).  It is important to ensure that the abstract is understandable, clear, and concise.

Response1: thank you. As per your suggestion, we have combined these two sentences – please see lines 14-17 of the tracked version of the manuscript and as copied below for your convenience:

"This comparative pre-post intervention study investigated the feasibility and benefits of Kegel exercises amongst incontinent women, prior to commencing resistance training (RT), to reduce the risk of stress urinary incontinence (SUI) compared to a group of women without prior Kegel exercises (KE)”

We have also included an explantion of the body composition (body mass index, fat and muscle mass) in line 19.

Point2:  Introduction – The first sentence of the introduction in line 31 and 32 says that “stress UI affects up to %70 of women” but related reference number 1 is saying that “the UI affects...” not stress UI. Additionally, the reference number 1 took this information from another reference which is 2016. In this case, is there more recent data or information from a direct resource? .

Response2thank you.  We have amended this and provided further clarity in this section – please see lines 33-37 of the tracked version of the manuscript which we have copied below for your convenience”

“Urinary incontinence affects up to 70% of women worldwide (1) with stress urinary incontinence (SUI) reported as the most prevalent sub-type (2). One study determined 40% of women in the U.S. experience SUI (3), a study in China considered almost 34% of SUI in a cohort of 4000 women to be high (4) and in a systematic review, prevalence of SUI varied from 12.5% up to 79% (5).”

Point3:  In the second paragraph of the introduction; * The first sentence in line 38 and 39 says “...some incontinent women...experienced...daily incontinence”, so what is “some”? How many percent of the women actually?

Response3  thank you. We have reviewed this further and agree that it needs to be clearer – you will note the amendment now in line 51 of the tracked version of the manuscript where we have indicated a figure of 8%.

Point4:  In the third paragraph of the introduction, the first sentence in line 46-47 is saying that “.....does not cause UI”. But who has it? The related resource says that ..”in overweight inactive woman”. That is why some parts like mentioned above are leaving questions unanswered.

Response4: thank you. With respect, the authors have looked at this and feel that it is clear and reads well in context.

Point5: In the line 53, IAP is mentioned but no long version is given (also not in previous parts).

Response5: thank you. We have amended this, please see line 62/63 of the tracked version of the manuscript which now reads:  “….when intra-abdominal pressure (IAP) is momentarily….”

Point6: This topic is interesting and probably this study is well done. However, in the way it is written many things have been overlooked and the connections are a little disconnected. Also, for example, it is causing curiosity while reading what resistance training is and what kind of activities it involves, and it is not explained.   

Response6: thank you. A brief explanation of resistance training (RT) has been provided in the introduction section to introduce the reader to what RT is – please see lines 41 – 44 of the tracked version of the manuscript. 

A descriptive of the RT that was utilised in the study is explained further in the Study Intervention section – specifically lines 150-151, 167-182.

Additionally, we feel that the table describing the example of the training program adds to this information providing concise information about the exercises and repetitions and is laid out clearly for replication of the study.

Point7: It is necessary to improve and construct the writing of the introduction by focusing on the topic and enrich it with the results of the studies. Some references are quite old, i.e. 1999, 2005 or 2006, so many things may have changed since then. It should be updated with more recent references wherever possible.  

Response7: Thank you. We have updated the older references as indicated and improved some of the content related to these also to provide more up to date information. We have kept a reference from 2006 (Madill et al.) as we found this to be an important study, however we have provided further support in another article by the same author and a systematic review. 

Please note that there is one additional reference from 1948 included, however this is a seminal work from Arnold Kegel (founder of Kegel exercises).

Of note, we found a paucity of research related specifically to stress urinary incontinence (SUI) that did not include pregnancy/parity – that is, studies about SUI generally, and therefore feel this is why our paper is important to further add to the general research on the topic.

We feel that in the intro and discussion we have included some further substantial points that add to the strength of this paper.

Point8:  Method. The authors provided details on the procedure and protocol of the study. The methods section was well written with clear and detailed objectives and descriptions.

Under the section Study Sample, it is written “this report compares the findings of two separate studies” in line 91. However it is mentioned about only one study which is the one KE+RT group (n=19) were recruited. But not a second study is mentioned. In this case, is the RT group (n=14) also recruited from a study?  Additionally, Why 14 and 19 women? How was the sample calculated? 

Response8: thank you. Please note that further into the paragraph it states “…states …..were recruited on completion of a successful published study [12]”, however we have attempted to make this clearer and refer to the figure of the summary of study protocol which also depicts this providing visual clarity. Please see paragraph of lines 108-118 of the tracked version of the manuscript.     

“This report compares the findings of two separate studies as described in figure 1 in which the effect of RT on SUI and PFMS was compared between participants who completed 12 weeks of RT with or without prior Kegel exercises. The RT only group consisted of 14 incontinent women who did not have any experience in RT or KE and completed 12 week of RT. Participants in the Kegel exercise plus resistance training (KE+RT) group (n=19) had completed 12 weeks of Kegel exercises under the supervision of a qualified pelvic floor specialist (15); they then completed the identical RT program as the RT group. A convenience sample was used in this study due to time requirements and availability of space and resources to carry out the project”

With regards to the sample number, this study used a convenience sample (this has been included as per above).

Point9:  Under the section Study Interventions, in line 102, “...safety for inclusion into the RT program was ascertained ....” but how? And what were the safety criterias you have ascertained?  

Response9: thank you.  The following has been added under the study procedure [lines 206-208 of the tracked version of the manuscript] – please see below for your convenience:

”A self-screening tool “physical activity readiness questionnaire (PAR-Q)” was required to rule out participants not suitable for physical exercise”.

This self-reported questionnaire guides the participant to whether they need to see their Doctor prior to inclusion in the program. No participant was excluded based on their response to this indicating they all were fit to participate.

Lines 200-205 of the tracked version of the manuscript outlines the inclusion/exclusion criteria met by the participants.

Point10:  A vital question: how was patient compliance with the management of the training and exercises program? How was it managed? 

Response10: thank you, yes this is vital. It is worth noting that we did not any issues with compliance as the participants were eager and enthusiastic to take part in this study. 

We have however included the following about compliance – please see lines 156-158 of the tracked version of the manuscript – we have included this below for your convenience:

“One-hour sessions were scheduled with the expectation of attendance twice per week to a maximum of twenty-four sessions. Compliance was managed with flexibility of multiple sessions to choose from, and attendance was documented by trainers along with notes made each session for continuation of training in next session. To ensure safety of participants with correct lifting technique, classes were limited to a maximum of six participants.”

Point11:   Results . Results are clearly explained. In its current form, results are easy to follow. The information in the lines 202-209 can be moved to method part.

Response11: With grateful thanks and respect for the time you have taken to review this paper, the authors have considered this feedback with due consideration however the general consensus was to keep this here as per the PRISMA reporting guidelines.

Point12: Also, the sociodemographic characteristics of the participants were not given.

Response12: thank you – we have added the following to address this under results – please see lines 228-229 of the tracked version of the manuscript. Also below for your convenience:

"A total of 24 participants aged between 27 and 62 years of age completed the study with an average age of 49.7 years. ….”

Point13:      One question is that have you considered that just as this patient mentioned in the results had exacerbation of arthritis, this study may also have some negative consequences in other individuals?

Response13: No other participant verbalised issues and all were enthusiastic about the project.

This particular participant was a fit and active 80 year old who travelled extensively. She wanted to participate and was fit however realised into the study it was beyond her capacities. She also needed to travel and that would also impact her time commitment.

Point 14:  Discussion In the discussion it is easy to follow the authors' direction when explaining different points. Clear and easy to understand writing with relevant information and important claims are pointed out.

The references to the claims could be increased (usually one reference is used??).  However, it is not in-depth enough and some conclusions are not adequately justified and discussed with sufficient evidence. More detail is needed on how Kegel supports your results and how this compares with other studies.

Response14: Thank you for this advice.

The consensus from authors is that the study design was only to preliminary explore this topic with the small sample. However, we have looked at some of this and added some further supporting references, added to the discussion generally and believe this now provides a more robust discussion and has improved the quality of this paper.

Point15:   The long version of “TrA” is not given in line 272 in the discussion.

Response15:     thank you.  We have amended this and included the full explanation - the transverse abdominis (TrA) – please see line 304 of the tracked version of the manuscript.

Limitations are sufficiently explained

Thank you reviewer 2 – we hope you appreciate the time we have taken to amend the paper incorporating your expert advice and believe we are now providing a better quality paper from this.

Reviewer 3 Report

Congratulations for your work, I think it's a valuable study that provides evidence with quantitative values. I have just a few suggestions:

Lines 62-67: Especially the last sentence of the authors in this paragraph is just an idea without proof and without any reference. The effect of kegel exercises on the pelvic floor muscles is already known. However, the mechanism was simplified by authors (You can explain better, not only by telling "it may be possible". )

Table 1 (also it is an detailed appendix of exercises) have to be given as an Appendix 1.

Author Response

Thank you reviewer 3 – we appreciate the time you have taken to review and provide valuable feedback on this paper and hope that we have addressed all of your concerns.  We look forward to a favourable response from you.

Congratulations for your work, I think it's a valuable study that provides evidence with quantitative values. I have just a few suggestions:

Point1         Lines 62-67: Especially the last sentence of the authors in this paragraph is just an idea without proof and without any reference. The effect of kegel exercises on the pelvic floor muscles is already known. However, the mechanism was simplified by authors (You can explain better, not only by telling "it may be possible". )

Response1    thank you. We have amended this section based on your feedback. It now reads, as per lines 75-84 in the tracked version of paper as:

"It is unfortunate that many sports are avoided due to SUI. In agreeance with a similar thought process related to continence threshold in another study (14), it is offered that the provocation of urinary incontinence in RT occurs as the result of inadequate strength of pelvic floor muscles, or the inability to correctly and timely activate them to counter the increase in intra-abdominal pressure that surpasses a woman’s individual continence capacity. A systematic review found that exercises which focused on the participant maintaining an awareness of PF muscle contractions, improved the efficacy of the activity (24); this aligns with the results of our study"

Point2:  Table 1 (also it is an detailed appendix of exercises) have to be given as an Appendix 1.

Response2: Thank you. We have given this deep consideration but the general consensus is that the authors are happy with its position and inclusion in the main paper as it provides a detailed regime of exercises which Industry would recognise.

We have made some overall improvements to the introduction and discussion and believe we are presenting you with a more robust paper which reflects the quality of this journal. We look forward to hearing from your with a favourable response.

Reviewer 4 Report

I regret to inform that I cannot recommend the publication of this article in this journal. The main reason is the design of the study. The authors have used two non-randomly divided groups, and the experimental group (KE+RT) consists of people who have been treated with supervised exercises and others with unsupervised exercises; data used in another publication where there were differences between both groups. So within that group (KE+RT) they did not start from the same conditions. It also does not indicate how much time has passed since the Kegel exercises intervention and the resistance training protocol, and if the RT group was monitored during the waiting time.

Furthermore, of all the variables presented, only two have been registered in clinicaltrials.gov (i.e., incontinence severity index (ISI) tool and pelvic floor muscle strength), lacking those referring to body composition.

Regarding the results, the inclusion of intragroup and intergroup comparisons using the Bonferroni correction is recommended. Thus, it would be possible to see more clearly the improvements of each group pre-post intervention and the differences between groups at each time measurement.

In addition, I consider that there is an overestimation of the results since both groups (KE+RT and RT) improved equally, which raises the question of whether the Kegel intervention before the RT program was really effective, or whether the indication of pelvic floor contraction during resistance exercises would be sufficient without exacerbating the UI.

Author Response

Dear reviewer 4 – thank you for your time and feedback. We are grateful for your review and provide the following response with great respect.

Point1    I regret to inform that I cannot recommend the publication of this article in this journal. The main reason is the design of the study. The authors have used two non-randomly divided groups, and the experimental group (KE+RT) consists of people who have been treated with supervised exercises and others with unsupervised exercises; data used in another publication where there were differences between both groups. So within that group (KE+RT) they did not start from the same conditions. It also does not indicate how much time has passed since the Kegel exercises intervention and the resistance training protocol, and if the RT group was monitored during the waiting time.

Response1: Please note The KE+RT group commenced the RT program immediately after the 12 weeks of Kegel exercise program. Both the RT group and the KE+RT group commenced the study with the same baseline condition – all having meeting inclusion/exclusion criteria.

The RT group did not need to be monitored baseline data was collected prior to the RT and then completion data.  The KE+RT group: baseline data was collected prior to the KE program and again on completion of the RT program.

Point2:   Furthermore, of all the variables presented, only two have been registered in clinicaltrials.gov (i.e., incontinence severity index (ISI) tool and pelvic floor muscle strength), lacking those referring to body composition.

Regarding the results, the inclusion of intragroup and intergroup comparisons using the Bonferroni correction is recommended. Thus, it would be possible to see more clearly the improvements of each group pre-post intervention and the differences between groups at each time measurement.

In addition, I consider that there is an overestimation of the results since both groups (KE+RT and RT) improved equally, which raises the question of whether the Kegel intervention before the RT program was really effective, or whether the indication of pelvic floor contraction during resistance exercises would be sufficient without exacerbating the UI.

Response2: We believe that we have addressed or justified some of your concerns in the Limitation section of the manuscript.

Additionally, We did consider a Bonferroni adjustment.  However, Bonferroni adjustments—and even modified variants such as the Holm procedure, the Holland-Copenhaver procedure, and the Hochberg procedure—are not recommended when the sample size is small.  They tend to be unduly conservative especially when the main hypothesis does not revolve around multiple comparisons.  Therefore, our original plan was to preclude Bonferroni adjustments. 

We agree that, because only ISI and pelvic floor muscle strength were registered, results associated with other variables need to be replicated before recommended in health interventions.

We agree that differences between the two groups—KE+RT and RT only—could, in principle, be ascribed to other pre-existing disparities these groups.  However, few pre-existing differences could readily explain the interaction effects. 

Dear Reviewer 4 - we believe we have made some changes to this paper which now provide a more robust paper. We believe this to be an important study which will add to the literature.

Round 2

Reviewer 4 Report

The authors have worked on the article and have given the necessary explanations to the previous reviewers' suggestions. However, some comments are described below:

While table 3 is orderly and all parameters are understood, table 2 looks messy. That is, the table footnote is missing. It is also not clear whether the data are mean and standard devitation or standard error, the meaning of the asterisks, the @80% value, among others. Regarding the asterisks, it is not known if they refer to time effect, condition (between subject effects) or interaction (time*group). This is important because there seems to be a mixture of significant results.

On the other hand, in table 3, a Ctrl image appears, which indicates that it is a screenshot. Please do not add the table as an image, and use the same typestyle. 

Please complete the Statistics section. It would be useful to explain Cohen's d values so that the reader can appreciate whether the effect size is small, medium or high. As well as the correlation values (low, medium or high). Explain whether the assumption of homoscedasticity using Levene's test and the sphericity using Mauchly's test were evaluated. Or why it has not been used.

It seems interesting to explain in the discussion why it is so difficult to make full attendance to all sessions (7 people dropped out for that reason). 

I consider grammatical corrections necessary. Some examples are: line 396 repeats "body mass index" followed by BMI; please, use only the abbreviation. The abbreviation IAP has not been described in the manuscript. In line 269 a sentence begins in lower case and the RT group is renamed to RT only group.

Author Response

Dear reviewer 4 - we are grateful for your further feedback and believe we have addressed it with due respect and consideration. We are very happy with how this paper has evolved.

The authors have worked on the article and have given the necessary explanations to the previous reviewers' suggestions. However, some comments are described below:

Point 1: While table 3 is orderly and all parameters are understood, table 2 looks messy. That is, the table footnote is missing. It is also not clear whether the data are mean and standard deviation or standard error, the meaning of the asterisks, the @80% value, among others. Regarding the asterisks, it is not known if they refer to time effect, condition (between subject effects) or interaction (time*group). This is important because there seems to be a mixture of significant results.

Response1: Thank you this table (line 239 of tracked version of the manuscript) has been amended.

  • The footnote has been re-inserted and this now clearly supports the table with clarity provided of information
  • Please note that title of this table indicates Mean data – we have attempted to make your feedback clearer in the title also – please see line 239 of the tracked version of the manuscript.
  • With regards to your query regarding time/condition/interaction, we believe this has been addressed in paragraphs from line 241 and incorporates data from table 3 further to support this explanation (and avoid duplication).

Point 2: On the other hand, in table 3, a Ctrl image appears, which indicates that it is a screenshot. Please do not add the table as an image, and use the same typestyle. 

Response2: Thank you.  We have reformatted this and it has now been inserted as a table and not image. This was initially done as it was very wide. Please find correctly formatted table. We have made some minor alterations to the table formatting and inserted a Note area in the footnote for further explanation to the reader. Please see from line 261 in the tracked version of the manuscript. We believe the table looks much clearer now – thank you for your valuable feedback.

Point 3: Please complete the Statistics section. It would be useful to explain Cohen's d values so that the reader can appreciate whether the effect size is small, medium or high. As well as the correlation values (low, medium or high). Explain whether the assumption of homoscedasticity using Levene's test and the sphericity using Mauchly's test were evaluated. Or why it has not been used.

Response3: Thank you.  After discussions with co-authors, we advise the Mauchly test of sphericity is not appliable when the repeated measures variable comprise only two levels.  In this instance, time comprises only two levels—before and after—obviating the need to conduct a Mauchly test of sphericity.

Regarding Cohen’s d vaues – Please note we have included in the footnote of table 2 a further explanation of this for the reader - this has been copied below for your convenience :

Note: Effect size of Cohen’s d value is based on d = small 0.2, medium effect 0.5 and large effect 0.8
Abbreviations: ISI - Incontinence Severity Index, PF - pelvic floor, MVC - maximum voluntary contraction time held in seconds, BMI - body mass index. *Significant difference from pre-intervention. 

Regarding your Pearsons correlation feedback - To provide further clarity around this, we have added the following line, as indicated in bold below, in the results section (please see line 257-258 of the tracked version of the manuscript):  

“A Pearson correlation, where 0 indicates no correlation and -1 indicates a perfect negative correlation, detected a statistically significant negative correlation between stress urinary incontinence and pelvic floor muscle strength, r(24)= -0.564, p= 0.004.”

Point 4: It seems interesting to explain in the discussion why it is so difficult to make full attendance to all sessions (7 people dropped out for that reason). 

Response 4: Thank you for your query. There were multiple reasons related to some of the participants not being able to attend the maximum of sessions required for data – these included common explanations such as unanticipated family and work commitments, illness and injuries unrelated to the training and transport issues. This did not lessen their enthusiasm for the study, only their time commitment for a brief period.

We have explained in the results section that “Data from 5 participants has been removed from the RT group and 2 further from the KE+RT group as they did not meet the requirement of attending 80% of the RT sessions".  

Point 5: I consider grammatical corrections necessary. Some examples are: line 396 repeats "body mass index" followed by BMI; please, use only the abbreviation. The abbreviation IAP has not been described in the manuscript. In line 269 a sentence begins in lower case and the RT group is renamed to RT only group.

Response5: Thank you – these corrections have been made in-text where they occurred.  I have made some grammatical corrections..

  • Line 398: removed body mass index and left abbreviation only
  • Line 415 : changed wording due to repetition of word “results”
  • RT Only group has now been changed to RT Group in lines 21, 261, 409.
  • Please note that the description of IAP was described in the manuscript at its first appearance in the paper at line 63-64 indicating intra-abdominal pressure, thereafter it has been referred to as IAP.

I would like to take this opportunity to thank you for a comprehensive look at this paper. We believe it to be greatly improved as a result and look forward to a favourable response from you.